# Towards Non-Asymptotic Convergence for Diffusion-Based Generative Models

**Gen Li**[*]    **Yuting Wei**[†]    **Yuxin Chen**[†‡]    **Yuejie Chi**[§]

## Abstract

Diffusion models, which convert noise into new data instances by learning to reverse a Markov diffusion process, have become a cornerstone in contemporary generative modeling. While their practical power has now been widely recognized, the theoretical underpinnings remain far from mature. In this work, we develop a suite of non-asymptotic theory towards understanding the data generation process of diffusion models in discrete time, assuming access to $\ell_2$-accurate estimates of the (Stein) score functions. For a popular deterministic sampler (based on the probability flow ODE), we establish a convergence rate proportional to $1/T$ (with $T$ the total number of steps), improving upon past results; for another mainstream stochastic sampler (i.e., a type of the denoising diffusion probabilistic model), we derive a convergence rate proportional to $1/\sqrt{T}$, matching the state-of-the-art theory. Imposing only minimal assumptions on the target data distribution (e.g., no smoothness assumption is imposed), our results characterize how $\ell_2$ score estimation errors affect the quality of the data generation process. In contrast to prior works, our theory is developed based on an elementary yet versatile non-asymptotic approach without resorting to toolboxes for SDEs and ODEs.

## 1 Introduction

Diffusion models have emerged as a cornerstone in contemporary generative modeling, a task that learns to generate new data instances (e.g., images, text, audio) that look similar in distribution to the training data (Ho et al., 2020; Sohl-Dickstein et al., 2015; Song & Ermon, 2019; Dhariwal & Nichol, 2021; Jolicoeur-Martineau et al., 2021; Chen et al., 2021; Kong et al., 2021; Austin et al., 2021). Originally proposed by Sohl-Dickstein et al. (2015) and later popularized by Song & Ermon (2019); Ho et al. (2020), the mainstream diffusion generative models — e.g., denoising diffusion probabilistic models (DDPMs) (Ho et al., 2020) and denoising diffusion implicit models (DDIMs) (Song et al., 2020a) — have underpinned major successes in content generators like DALL·E 2 (Ramesh et al., 2022), Stable Diffusion (Rombach et al., 2022) and Imagen (Saharia et al., 2022), claiming state-of-the-art performance in the now broad field of generative artificial intelligence (AI). See Yang et al. (2022); Croitoru et al. (2023) for overviews of recent development.

In a nutshell, a diffusion generative model is based upon two stochastic processes in $\mathbb{R}^d$:

1)  a forward process
$$X_0 \rightarrow X_1 \rightarrow \cdots \rightarrow X_T \tag{1}$$
that starts from a sample drawn from the target data distribution (e.g., of natural images) and gradually diffuses it into a noise-like distribution (e.g., standard Gaussians);

2)  a reverse process
$$Y_T \rightarrow Y_{T-1} \rightarrow \cdots \rightarrow Y_0 \tag{2}$$

---

[*]Department of Statistics, The Chinese University of Hong Kong, Hong Kong.

[†]Department of Statistics and Data Science, Wharton School, University of Pennsylvania, Philadelphia, PA 19104, USA.

[‡]Department of Electrical and Systems Engineering, University of Pennsylvania, Philadelphia, PA 19104, USA.

[§]Department of Electrical and Computer Engineering, Carnegie Mellon University, Pittsburgh, PA 15213, USA.

that starts from pure noise (e.g., standard Gaussians) and successively converts it into new samples sharing similar distributions as the target data distribution.

Transforming data into noise in the forward process is straightforward, often hand-crafted by increasingly injecting more noise into the data at hand. What is challenging is the construction of the reverse process: how to generate the desired information out of pure noise? To do so, a diffusion model learns to build a reverse process (2) that imitates the dynamics of the forward process (1) in a time-reverse fashion; more precisely, the design goal is to ascertain distributional proximity[1]

$$Y_t \overset{\mathrm{d}}{\approx} X_t, \qquad t = T, \cdots, 1 \tag{3}$$

through proper learning based on how the training data propagate in the forward process. Encouragingly, there often exist feasible strategies to achieve this goal as long as faithful estimates about the (Stein) score functions — the gradients of the log marginal density of the forward process — are available, an intriguing fact that can be illuminated by the existence and construction of reverse-time stochastic differential equations (SDEs) (Anderson, 1982; Haussmann & Pardoux, 1986) (see Section 2.2 for more precise discussions). Viewed in this light, a diverse array of diffusion models are frequently referred to as *score-based generative modeling (SGM)*. The popularity of SGM was initially motivated by, and has since further inspired, numerous recent studies on the problem of learning score functions, a subroutine that also goes by the name of score matching (e.g., Hyvärinen (2005; 2007); Vincent (2011); Song et al. (2020b); Koehler et al. (2023)).

Nonetheless, despite the mind-blowing empirical advances, a mathematical theory for diffusion generative models is still in its infancy. Given the complexity of developing a full-fledged end-to-end theory, a divide-and-conquer approach has been advertised, decoupling the score learning phase (i.e., how to estimate score functions from training data) and the generative sampling phase (i.e., how to generate new data given the score estimates). In particular, the past two years have witnessed growing interest and remarkable progress from the theoretical community towards understanding the sampling phase (Block et al., 2020; De Bortoli et al., 2021; Liu et al., 2022; De Bortoli, 2022; Lee et al., 2023; Pidstrigach, 2022; Chen et al., 2022b;a; Tang, 2023; Chen et al., 2023c; Tang & Zhao, 2024; Li et al., 2024a). For instance, polynomial-time convergence guarantees have been established for stochastic samplers (e.g., Chen et al. (2022b;a); Benton et al. (2023a)) and deterministic samplers (e.g., Chen et al. (2023c); Benton et al. (2023b)), both of which accommodated a fairly general family of data distributions.

**This paper.** The present paper contributes to this growing list of theoretical endeavors by developing a new suite of non-asymptotic theory for several score-based generative modeling algorithms. We concentrate on two types of samplers (Song et al., 2021b) in discrete time: (i) a deterministic sampler based on a sort of ordinary differential equations (ODEs) called probability flow ODEs (which is closely related to the DDIM); and (ii) a DDPM-type stochastic sampler motivated by reverse-time SDEs. We impose only minimal assumptions on the target data distribution (e.g., no smoothness condition is needed), and would like to quantify the impact of $\ell_2$ score estimation errors. In comparisons to past works, our main contributions are three-fold.

*Non-asymptotic convergence guarantees.* For a popular deterministic sampler, we demonstrate that the number of steps needed to yield $\varepsilon$-accuracy — meaning that the total variation (TV) distance between the distribution of $X_1$ and that of $Y_1$ is no larger than $\varepsilon$ — is proportional to $1/\varepsilon$ (in addition to other polynomial dimension dependency). This improves upon prior convergence guarantees (Chen et al., 2023c) and does not exhibit exponential dependency on the smoothness assumption as in Chen et al. (2023c); Benton et al. (2023b). For another DDPM-type stochastic sampler, we establish an iteration complexity proportional to $1/\varepsilon^2$, matching existing theory Chen et al. (2022b;a); Benton et al. (2023a) in terms of the $\varepsilon$-dependency.

*Score estimation errors for the determinstic sampler.* In our convergence guarantees for the deterministic sampler, the TV distance between $X_1$ and $Y_1$ are shown to be proportional to the $\ell_2$ score estimation error as well as the associated Jacobian errors. As far as we know, this is the first result for this deterministic sampler that accounts for score estimation errors in discrete time. In comparison, other theoretical results that accommodate score errors for the probability flow ODE approach

---

[1]Two random vectors $X$ and $Y$ are said to obey $X \overset{\mathrm{d}}{=} Y$ (resp. $X \overset{\mathrm{d}}{\approx} Y$) if they are equivalent (resp. close) in distribution.

either study certain stochastic variations of this deterministic sampler (Chen et al., 2023b) or fall short of accommodating discretization errors (Benton et al., 2023b).

*An elementary non-asymptotic analysis framework.* From the technical viewpoint, the analysis framework laid out in this paper is fully non-asymptotic in nature. In contrast to prior analyses that take a detour to study the continuum limits and then control the discretization error, our approach tackles the discrete-time processes directly using elementary analysis strategies. No knowledge of SDEs or ODEs is required for establishing our theory, thereby resulting in a more versatile framework and sometimes lowering the technical barrier towards understanding diffusion models.

**Notation.** For any two functions $f(d,T)$ and $g(d,T)$, we adopt the notation $f(d,T) \lesssim g(d,T)$ or $f(d,T) = O(g(d,T))$ (resp. $f(d,T) \gtrsim g(d,T)$) to mean that there exists some universal constant $C_1 > 0$ such that $f(d,T) \leq C_1 g(d,T)$ (resp. $f(d,T) \geq C_1 g(d,T)$) for all $d$ and $T$; moreover, the notation $f(d,T) \asymp g(d,T)$ indicates that $f(d,T) \lesssim g(d,T)$ and $f(d,T) \gtrsim g(d,T)$ hold at once. The notation $\widetilde{O}(\cdot)$ is defined similar to $O(\cdot)$ except that it hides the logarithmic dependency. Additionally, the notation $f(d,T) = o(g(d,T))$ means that $f(d,T)/g(d,T) \to 0$ as $d, T$ tend to infinity. For any two probability measures $P$ and $Q$, the total variation (TV) distance between them is defined to be $\mathsf{TV}(P,Q) := \frac{1}{2} \int |\mathrm{d}P - \mathrm{d}Q|$. Throughout the paper, $p_X(\cdot)$ (resp. $p_{X\,|\,Y}(\cdot\,|\,\cdot)$) denotes the probability density function of $X$ (resp. $X$ given $Y$). For any matrix $A$, we denote by $\|A\|$ (resp. $\|A\|_{\mathrm{F}}$) the spectral norm (resp. Frobenius norm) of $A$. Also, for any vector-valued function $f$, we let $J_f$ or $\frac{\partial f}{\partial x}$ represent the Jacobian matrix of $f$.

## 2 PRELIMINARIES

In this section, we introduce the basics of diffusion generative models. The ultimate goal of a generative model can be concisely stated: given data samples drawn from an unknown distribution of interest $p_{\mathsf{data}}$ in $\mathbb{R}^d$, we wish to generate new samples whose distributions closely resemble $p_{\mathsf{data}}$.

### 2.1 DIFFUSION GENERATIVE MODELS

Towards achieving the above goal, a diffusion generative model typically encompasses two Markov processes: a forward process and a reverse process, as described below.

**The forward process.** In the forward chain, one progressively injects noise into the data samples to diffuse and obscure the data. The distributions of the injected noise are often hand-picked, with the standard Gaussian distribution receiving widespread adoption. Specifically, the forward Markov process produces a sequence of $d$-dimensional random vectors $X_1 \to X_2 \to \cdots \to X_T$ as follows:

$$X_0 \sim p_{\mathsf{data}}, \tag{4a}$$

$$X_t = \sqrt{1 - \beta_t}\, X_{t-1} + \sqrt{\beta_t}\, W_t, \qquad 1 \leq t \leq T, \tag{4b}$$

where $\{W_t\}_{1 \leq t \leq T}$ indicates a sequence of independent noise vectors drawn from $W_t \overset{\text{i.i.d.}}{\sim} \mathcal{N}(0, I_d)$. The hyper-parameters $\{\beta_t \in (0,1)\}$ represent prescribed learning rate schedules that control the variance of the noise injected in each step. If we define

$$\alpha_t := 1 - \beta_t, \qquad \overline{\alpha}_t := \prod_{k=1}^{t} \alpha_k, \qquad 1 \leq t \leq T, \tag{5}$$

then it can be straightforwardly verified that for every $1 \leq t \leq T$,

$$X_t = \sqrt{\overline{\alpha}_t}\, X_0 + \sqrt{1 - \overline{\alpha}_t}\, \overline{W}_t \qquad \text{for some } \overline{W}_t \sim \mathcal{N}(0, I_d). \tag{6}$$

Clearly, if the covariance of $X_0$ is also equal to $I_d$, then the covariance of $X_t$ is preserved throughout the forward process; for this reason, this forward process (4) is sometimes referred to as *variance-preserving* (Song et al., 2021b). Throughout this paper, we employ the notation

$$q_t := \mathsf{law}(X_t) \tag{7}$$

to denote the distribution of $X_t$. As long as $\overline{\alpha}_T$ is vanishingly small, one has the following property for a general family of data distributions:

$$q_T \approx \mathcal{N}(0, I_d). \tag{8}$$

**The reverse process.** The reverse chain $Y_T \to Y_{T-1} \to \ldots \to Y_1$ is designed to (approximately) revert the forward process, allowing one to transform pure noise into new samples with matching distributions as the original data. To be more precise, by initializing it as

$$Y_T \sim \mathcal{N}(0, I_d), \tag{9a}$$

we seek to design a reverse-time Markov process with nearly identical marginals as the forward process, namely,

$$\text{(goal)} \qquad Y_t \overset{\text{d}}{\approx} X_t, \qquad t = T, T-1, \cdots, 1. \tag{9b}$$

Throughout the paper, we often employ the following notation to indicate the distribution of $Y_t$:

$$p_t := \mathsf{law}(Y_t). \tag{10}$$

## 2.2 DETERMINISTIC VS. STOCHASTIC SAMPLERS: A CONTINUOUS-TIME INTERPRETATION

Evidently, the most crucial step of the diffusion model lies in effective design of the reverse process. Two mainstream approaches stand out:

- *Deterministic samplers.* Starting from $Y_T \sim \mathcal{N}(0, I_d)$, this approach selects a set of functions $\{\Phi_t(\cdot)\}_{1 \le t \le T}$ and computes:

  $$Y_{t-1} = \Phi_t(Y_t), \qquad t = T, \cdots, 1. \tag{11}$$

  Clearly, the sampling process is fully deterministic except for the initialization $Y_T$.

- *Stochastic samplers.* Initialized again at $Y_T \sim \mathcal{N}(0, I_d)$, this approach computes another collection of functions $\{\Psi_t(\cdot, \cdot)\}_{1 \le t \le T}$ and performs the updates:

  $$Y_{t-1} = \Psi_t(Y_t, Z_t), \qquad t = T, \cdots, 1, \tag{12}$$

  where the $Z_t$'s are independent noise vectors obeying $Z_t \overset{\text{i.i.d.}}{\sim} \mathcal{N}(0, I_d)$.

In order to elucidate the feasibility of the above two approaches, we find it helpful to look at the continuum limit through the lens of SDEs and ODEs. It is worth emphasizing, however, that the development of our main theory does *not* rely on any knowledge of SDEs and ODEs.

- *The forward process.* A continuous-time analog of the forward process can be modeled as

  $$\mathrm{d}X_t = f(X_t, t)\mathrm{d}t + g(t)\mathrm{d}W_t \quad (0 \le t \le T), \qquad X_0 \sim p_{\mathsf{data}} \tag{13}$$

  for some functions $f(\cdot, \cdot)$ and $g(\cdot)$ (denoting respectively the drift and diffusion coefficient), where $W_t$ denotes a $d$-dimensional standard Brownian motion. As a special example, the continuum limit of (4) takes the following form[2] (Song et al., 2021b)

  $$\mathrm{d}X_t = -\frac{1}{2}\beta(t)X_t\mathrm{d}t + \sqrt{\beta(t)}\,\mathrm{d}W_t \quad (0 \le t \le T), \qquad X_0 \sim p_{\mathsf{data}} \tag{14}$$

  for some function $\beta(t)$. As before, we denote by $q_t$ the distribution of $X_t$ in (13).

- *The reverse process.* As it turns out, the following two reverse processes are both capable of reconstructing the distribution of the forward process, motivating the design of two distinctive samplers. Here and throughout, we use $\nabla \log q_t(X)$ to abbreviate $\nabla_X \log q_t(X)$ for notational simplicity.

  - One feasible approach is to the so-called *probability flow ODE* (Song et al., 2021b)

    $$\mathrm{d}Y_t^{\mathsf{ode}} = \left( -f(Y_t^{\mathsf{ode}}, T-t) + \frac{1}{2}g(T-t)^2 \nabla \log q_{T-t}(Y_t^{\mathsf{ode}}) \right)\mathrm{d}t \quad (0 \le t \le T), \tag{15}$$

    with $Y_0^{\mathsf{ode}} \sim q_T$, which exhibits matching distributions as follows:

    $$Y_{T-t}^{\mathsf{ode}} \overset{\text{d}}{=} X_t, \qquad 0 \le t \le T.$$

    The deterministic nature of this approach often enables faster sampling. It has been shown that this family of deterministic samplers is closely related to the DDIM sampler (Karras et al., 2022; Song et al., 2021b).

---

[2]To see its connection with (4), it suffices to derive from (4) that $X_t - X_{t-\mathrm{d}t} = \sqrt{1-\beta_t}X_{t-\mathrm{d}t} - X_{t-\mathrm{d}t} + \sqrt{\beta_t}W_t \approx -\frac{1}{2}\beta_t X_{t-\mathrm{d}t} + \sqrt{\beta_t}W_t.$

– In view of the classical results Anderson (1982); Haussmann & Pardoux (1986), one can also construct a "reverse-time" SDE

$$\mathrm{d}Y_t^{\mathsf{sde}} = \Big( -f\big(Y_t^{\mathsf{sde}}, T-t\big) + g(T-t)^2 \nabla \log q_{T-t}\big(Y_t^{\mathsf{sde}}\big) \Big)\mathrm{d}t + g(T-t)\mathrm{d}Z_t^{\mathsf{sde}} \tag{16}$$

for $0 \leq t \leq T$, with $Y_0^{\mathsf{sde}} \sim q_T$ and $Z_t^{\mathsf{sde}}$ being a standard Brownian motion. Strikingly, this process also satisfies

$$Y_{T-t}^{\mathsf{sde}} \overset{\mathrm{d}}{=} X_t, \qquad 0 \leq t \leq T.$$

The popular DDPM sampler (Ho et al., 2020) falls under this category.

Interestingly, in addition to the functions $f$ and $g$ that define the forward process, construction of both (15) and (16) relies only upon the knowledge of the gradient of the log density $\nabla \log q_t(\cdot)$ of the intermediate steps of the forward diffusion process — often referred to as the (Stein) score function. Consequently, a key enabler of the above paradigms lies in reliable learning of the score function, and hence the name *score-based generative modeling*.

## 3 Algorithms and main results

In this section, we analyze a couple of diffusion generative models, including both deterministic and stochastic samplers. While the proofs for our main theory are all postponed to the appendix, it is worth emphasizing upfront that our analysis framework directly tackles the discrete-time processes without resorting to any toolbox of SDEs and ODEs tailored to the continuous-time limits. This elementary approach might potentially be versatile for analyzing a broad class of variations of these samplers. For instance, prior ODE-based theory (e.g., Chen et al. (2023b;c)) encountered certain technical challenges when analyzing the deterministic sampler directly, and our elementary approach is able to shed new light on the convergence of this important sampler.

### 3.1 Assumptions and learning rates

Before proceeding, we impose some assumptions on the score estimates and the target data distributions, and specify the hypter-parameters $\{\alpha_t\}$, which shall be adopted throughout all cases.

**Score estimates.** Given that the score functions are an essential component in score-based generative modeling, we assume access to faithful estimates of the score functions $\nabla \log q_t(\cdot)$ across all intermediate steps $t$, thus disentangling the score learning phase and the data generation phase. Towards this end, let us first formally introduce the true score function as follows.

**Definition 1** (Score function). *The score function, denoted by $s_t^\star : \mathbb{R}^d \to \mathbb{R}^d$, is defined as*

$$s_t^\star(X) := \nabla \log q_t(X), \qquad 1 \leq t \leq T. \tag{17}$$

As has been pointed out by previous works concerning score matching (e.g., Hyvärinen (2005); Vincent (2011); Chen et al. (2022b)), the score function $s_t^\star$ admits an alternative form as follows (owing to properties of Gaussian distributions):

$$s_t^\star := \arg \min_{s:\mathbb{R}^d \to \mathbb{R}^d} \mathop{\mathbb{E}}_{W \sim \mathcal{N}(0,I_d), X_0 \sim p_{\mathsf{data}}} \left[ \left\| s\big(\sqrt{\overline{\alpha}_t}X_0 + \sqrt{1-\overline{\alpha}_t}W\big) + \frac{1}{\sqrt{1-\overline{\alpha}_t}}W \right\|_2^2 \right], \tag{18}$$

which takes the form of the minimum mean square error estimator for $-\frac{1}{\sqrt{1-\overline{\alpha}_t}}W$ given $\sqrt{\overline{\alpha}_t}X_0 + \sqrt{1-\overline{\alpha}_t}W$ and is often more amenable to training.

With Definition 1 in place, we can readily introduce the following assumptions that capture the quality of the score estimate $\{s_t\}_{1 \leq t \leq T}$ we have available.

**Assumption 1.** *Suppose that the score function estimate $\{s_t\}_{1 \leq t \leq T}$ obeys*

$$\frac{1}{T}\sum_{t=1}^T \mathop{\mathbb{E}}_{X \sim q_t} \left[ \big\| s_t(X) - s_t^\star(X) \big\|_2^2 \right] \leq \varepsilon_{\mathsf{score}}^2. \tag{19}$$

**Assumption 2.** *For each $1 \leq t \leq T$, assume that $s_t(\cdot)$ is continuously differentiable, and denote by $J_{s_t^\star} = \frac{\partial s_t^\star}{\partial x}$ and $J_{s_t} = \frac{\partial s_t}{\partial x}$ the Jacobian matrices of $s_t^\star(\cdot)$ and $s_t(\cdot)$, respectively. Assume that the score function estimate $\{s_t\}_{1 \leq t \leq T}$ obeys*

$$\frac{1}{T}\sum_{t=1}^{T} \mathop{\mathbb{E}}_{X \sim q_t} \left[ \left\| J_{s_t}(X) - J_{s_t^\star}(X) \right\| \right] \leq \varepsilon_{\mathsf{Jacobi}}. \tag{20}$$

In a nutshell, Assumption 1 reflects the $\ell_2$ score estimation error, whereas Assumption 2 concerns the estimation error in terms of the corresponding Jacobian matrix. Both assumptions consider the *average* estimation errors over all $T$ steps. Our theory for the deterministic sampler relies on both Assumptions 1 and 2, while the theory for the stochastic sampler requires only Assumption 1. We shall discuss in Section 3.2 the insufficiency of Assumption 1 alone for the deterministic sampler.

**Target data distributions.** Our goal is to uncover the effectiveness of diffusion models in generating a broad family of data distributions. Throughout this paper, the only assumptions we need to impose on the target data distribution $p_{\mathsf{data}}$ are the following:

- $X_0$ is an absolutely continuous random vector, and
$$\mathbb{P}\big(\|X_0\|_2 \leq T^{c_R} \mid X_0 \sim p_{\mathsf{data}}\big) = 1 \tag{21}$$
for some arbitrarily large constant $c_R > 0$.

This assumption allows the radius of the support of $p_{\mathsf{data}}$ to be exceedingly large (given that the exponent $c_R$ can be arbitrarily large).

**Learning rate schedule.** Let us also take a moment to specify the learning rates to be used for our theory and analyses. For some large enough numerical constants $c_0, c_1 > 0$, we set

$$\beta_1 = 1 - \alpha_1 = \frac{1}{T^{c_0}}; \tag{22a}$$

$$\beta_t = 1 - \alpha_t = \frac{c_1 \log T}{T} \min \left\{ \beta_1 \Big(1 + \frac{c_1 \log T}{T}\Big)^t, 1 \right\}. \tag{22b}$$

**Remark 1.** *As we shall see in our analysis, the discretization error depends crucially upon the quantity $\frac{1-\alpha_t}{1-\overline{\alpha}_t}$, whereas the initialization error relies heavily upon $\overline{\alpha}_1$ and $\overline{\alpha}_T$. Based on these observations, our learning rate schedule (22) is designed to make $\frac{1-\alpha_t}{1-\overline{\alpha}_t}$ as small as possible, while making sure $\overline{\alpha}_1$ (resp. $\overline{\alpha}_T$) is close to 1 (resp. 0). These properties will be shown in (43). Moreover, it is worth noting that our analysis can be easily extended to accommodate a much broader class of learning rates, although the resulting convergence rates might vary.*

## 3.2 An ODE-based deterministic sampler

We begin by analyzing a deterministic sampler: a discrete-time version of the probability flow ODE.

Armed with the score estimates $\{s_t\}_{1 \leq t \leq T}$, a discrete-time version of the probability flow ODE approach (cf. (15)) adopts the following update rule:

$$Y_T \sim \mathcal{N}(0, I_d), \qquad Y_{t-1} = \Phi_t\big(Y_t\big) \quad \text{for } t = T, \cdots, 1, \tag{23a}$$

where $\Phi_t(\cdot)$ is taken to be

$$\Phi_t(x) := \frac{1}{\sqrt{\alpha_t}}\left( x + \frac{1 - \alpha_t}{2} s_t(x) \right). \tag{23b}$$

This approach, based on the probability flow ODE (15), often achieves faster sampling compared to the stochastic counterpart (Song et al., 2021b). Despite the empirical advances, however, the theoretical understanding of this type of deterministic samplers remained far from mature.

We first derive non-asymptotic convergence guarantees — measured by the total variation distance between the forward and the reverse processes — for the above deterministic sampler (23). The proof of this result can be found in Li et al. (2023, Section 5.2).

**Theorem 1.** *Suppose that* (21) *holds true. Assume that the score estimates* $s_t(\cdot)$ $(1 \leq t \leq T)$ *satisfy Assumptions 1 and 2. Then the sampling process* (23) *with the learning rate schedule* (22) *satisfies*

$$\mathsf{TV}(q_1, p_1) \leq C_1 \frac{d^2 \log^4 T}{T} + C_1 \frac{d^6 \log^6 T}{T^2} + C_1 \sqrt{d \log^3 T} \varepsilon_{\mathsf{score}} + C_1 d(\log T) \varepsilon_{\mathsf{Jacobi}} \qquad (24)$$

*for some universal constants* $C_1 > 0$, *where* $p_1$ *(resp.* $q_1$*) represents the distribution of* $Y_1$ *(resp.* $X_1$*).*

**Implications.** Let us highlight the main implications of Theorem 1. Before proceeding, note that our theory is concerned with convergence to $q_1$. Given that $X_1 \sim q_1$ and $X_0 \sim q_0$ are very close due to the choice of $\alpha_1$, focusing on the convergence w.r.t. $q_1$ instead of $q_0$ remains practically relevant.

*(a) Iteration complexity.* Consider first the scenario that has access to perfect score estimates (i.e., $\varepsilon_{\mathsf{score}} = 0$). In order to achieve $\varepsilon$-accuracy (in the sense that $\mathsf{TV}(q_1, p_1) \leq \varepsilon$), the number of steps $T$ only needs to exceed

$$\widetilde{O}\big(d^2/\varepsilon + d^3/\sqrt{\varepsilon}\big). \qquad (25)$$

*(b) Stability.* Turning to the more general case with imperfect score estimates (i.e., $\varepsilon_{\mathsf{score}} > 0$), the deterministic sampler (23) yields a distribution whose distance to the target distribution (measured again by the TV distance) scales proportionally with $\varepsilon_{\mathsf{score}}$ and $\varepsilon_{\mathsf{Jacobi}}$. It is noteworthy that in addition to the score estimation errors, we are in need of an assumption on the stability of the associated Jacobian matrices, which plays a pivotal in ensuring that the reverse-time deterministic process does not deviate considerably from the desired process.

*(c) Insufficiency of the score estimation error assumption alone.* The careful reader might wonder why we are in need of additional assumptions beyond the $\ell_2$ score error stated in Assumption 1. To answer this question, we find it helpful to look at a simple example below.

- **Example.** Consider the case where $X_0 \sim \mathcal{N}(0, 1)$, and hence $X_1 \sim \mathcal{N}(0, 1)$. Suppose that the reverse process for time $t = 2$ can lead to the desired distribution if exact score function is employed, namely,

$$Y_1^\star := \frac{1}{\sqrt{\alpha_2}} \left( Y_2 - \frac{1 - \alpha_2}{2} s_2^\star(Y_2) \right) \sim \mathcal{N}(0, 1).$$

Now, suppose that the score estimate $s_2(\cdot)$ we have available obeys

$$s_2(y_2) = s_2^\star(y_2) + \frac{2\sqrt{\alpha_2}}{1 - \alpha_2} \left\{ y_1^\star - L \left\lfloor \frac{y_1^\star}{L} \right\rfloor \right\} \quad \text{with } y_1^\star := \frac{1}{\sqrt{\alpha_2}} \left( y_2 - \frac{1 - \alpha_2}{2} s_2^\star(y_2) \right)$$

for $L > 0$, where $\lfloor z \rfloor$ is the greatest integer not exceeding $z$. It follows that

$$Y_1 = Y_1^\star + \frac{1 - \alpha_2}{2\sqrt{\alpha_2}} \big[ s_2^\star(Y_2) - s_2(Y_2) \big] = L \left\lfloor \frac{Y_1^\star}{L} \right\rfloor.$$

Clearly, the score error $\mathbb{E}_{X_2 \sim \mathcal{N}(0,1)} \big[ |s_2(X_2) - s_2^\star(X_2)|^2 \big]$ can be made arbitrarily small by taking $L \to 0$. However, the discrete nature of $Y_1$ forces $\mathsf{TV}(Y_1, X_1) = 1$.

This example demonstrates that, for the deterministic sampler, the TV distance between $Y_1$ and $X_1$ might not improve as the score error decreases. If we wish to relax Assumption 2, one potential way is to resort to other metrics (e.g., Wasserstein distance) instead of TV distance between $Y_1$ and $X_1$.

*(d) Relaxing the boundedness assumption on $X_0$.* As it turns out, the assumption (21) can also be relaxed. Supposing that $\mathbb{P}\big(\|X_0\|_2 \leq B \mid X_0 \sim p_{\mathsf{data}}\big) = 1$ for some quantity $B > 0$ (which is allowed to grow faster than a polynomial in $T$), we can readily extend our analysis to obtain

$$\mathsf{TV}(q_1, p_1) \leq C_1 \frac{d^2 \log^4 T \log^2 B}{T} + C_1 \frac{d^6 \log^6 T \log^3 B}{T^2} + C_1 \sqrt{d \log^3 T \log B} \varepsilon_{\mathsf{score}} + C_1 d(\log T) \varepsilon_{\mathsf{Jacobi}}.$$

Importantly, the convergence rate depends only logarithmically in $B$.

**Comparisons with past works.** To the best of our knowledge, the only non-asymptotic analysis for the discretized probability flow ODE approach in prior literature was derived by a very recent work Chen et al. (2023c), which established the first non-asymptotic convergence guarantees that exhibit

polynomial dependency in both $d$ and $1/\varepsilon$ (see, e.g., Chen et al. (2023c, Theorem 4.1)). However, it fell short of providing concrete polynomial dependency in $d$ and $1/\varepsilon$, suffered from exponential dependency in the Lipschitz constant of the score function, and relied on exact score estimates. In contrast, our result in Theorem 1 uncovers a concrete $d^2/\varepsilon$ scaling (ignoring lower-order and logarithmic terms) without imposing any smoothness assumption on the target data distribution, and makes explicit the effect of score estimation errors, both which were previously unavailable for such discrete-time deterministic samplers. Another recent work Benton et al. (2023b) studied the convergence of the probability flow ODE approach without accounting for the discretization error; the result therein also exhibited exponential dependency on the Lipschitz constant. Finally, while we were wrapping up the current paper, we became aware of the independent work Chen et al. (2023b) establishing improved polynomial dependency for two variants of the probability flow ODE. By inserting an additional stochastic corrector step — based on overdamped (resp. underdamped) Langevin diffusion — in each iteration of the probability flow ODE (so strictly speaking, these variations are no longer deterministic samplers), Chen et al. (2023b) showed that $\widetilde{O}(L^3 d/\varepsilon^2)$ (resp. $\widetilde{O}(L^2\sqrt{d}/\varepsilon)$) steps are sufficient, where $L$ denotes the Lipschitz constant of the score function. In comparison, our result demonstrates for the first time that the plain probability flow ODE already achieves the $1/\varepsilon$ scaling without requiring either a corrector step; one limitation of our result, however, is the sub-optimal $d$-dependency compared to the variants studied in Chen et al. (2023b).

### 3.3 A DDPM-TYPE STOCHASTIC SAMPLER

Armed with the score estimates $\{s_t\}$, we can readily introduce the following stochastic sampler that operates in discrete time, motivated by the reverse-time SDE (16):

$$Y_T \sim \mathcal{N}(0, I_d), \qquad Y_{t-1} = \Psi_t(Y_t, Z_t) \quad \text{for } t = T, \cdots, 1 \tag{26a}$$

where $Z_t \overset{\text{i.i.d.}}{\sim} \mathcal{N}(0, I_d)$, and

$$\Psi_t(y, z) = \frac{1}{\sqrt{\alpha_t}}\Big(y + (1 - \alpha_t)s_t(y)\Big) + \sigma_t z \qquad \text{with } \sigma_t^2 = \frac{1}{\alpha_t} - 1. \tag{26b}$$

The key difference between this sampler and the deterministic sampler (23) is that: (i) there exists an additional pre-factor of $1/2$ on $s_t$ in the deterministic sampler; and (ii) the stochastic sampler injects additional noise $Z_t$ in each step.

In contrast to deterministic samplers, the stochastic samplers have received more theoretical attention, with the state-of-the-art results established by Chen et al. (2022b;a) as well as a very recent paper Benton et al. (2023a). The elementary approach developed in the current paper is also applicable towards understanding this type of samplers, leading to the following non-asymptotic theory.

**Theorem 2.** *Suppose* (21) *holds true. Equipped with the estimates in Assumption 1 and the learning rate schedule* (22)*, the stochastic sampler* (26) *achieves, for some universal constants $C_1 > 0$,*

$$\mathsf{TV}\big(q_1, p_1\big) \leq \sqrt{\frac{1}{2}\mathsf{KL}\big(q_1 \parallel p_1\big)} \leq C_1 \frac{d^2 \log^3 T}{\sqrt{T}} + C_1\sqrt{d}\varepsilon_{\mathsf{score}} \log^2 T. \tag{27}$$

Theorem 2 establishes non-asymptotic convergence guarantees for the stochastic sampler (26). As asserted by the theorem, if we have access to perfect score estimates, then the number of steps needed to attain $\varepsilon$-accuracy (measured by the TV distance between $p_1$ and $q_1$) is proportional to $1/\varepsilon^2$, matching the state-of-the-art $\varepsilon$-dependency derived in Chen et al. (2022a), albeit exhibiting a worse dimensional dependency. In addition, in the presence of score estimation error, the sampler achieves a TV distance proportional to $\varepsilon_{\mathsf{score}}$, again consistent with prior results. Our analysis follows a completely different path compared with the SDE-based approach in Chen et al. (2022a), thus offering complementary interpretations for this important sampler.

## 4 OTHER RELATED WORKS

**Theory for SGMs.** Early theoretical efforts in understanding the convergence of score-based stochastic samplers suffered from being either not quantitative (De Bortoli et al., 2021; Liu et al., 2022; Pidstrigach, 2022), or the curse of dimensionality (e.g., exponential dependencies in the convergence guarantees) (Block et al., 2020; De Bortoli, 2022). Lee et al. (2022) provided the first

polynomial convergence guarantees with $L_2$-accurate score estimates, for any smooth distribution satisfying the log-Sobolev inequality. Chen et al. (2022b); Lee et al. (2023); Chen et al. (2022a) subsequently lifted such a stringent data distribution assumption. More concretely, Chen et al. (2022b) accommodated a broad family of data distributions under the premise that the score functions over the entire trajectory of the forward process are Lipschitz; Lee et al. (2023) only required certain smoothness assumptions but came with worse dependence on the problem parameters; and more recent results in Chen et al. (2022a) applied to literally any data distribution with bounded second-order moment. In addition, Wibisono & Yang (2022) also established a convergence theory for score-based generative models, assuming that the error of the score estimator has a bounded moment generating function and that the data distribution satisfies the log-Sobolev inequality. Turning attention to samplers based on the probability flow ODE, Chen et al. (2023c) derived the first non-asymptotic bounds for this type of samplers. Improved convergence guarantees have recently been provided by a concurrent work Chen et al. (2023b), with the assistance of additional corrector steps inerspersed in each iteration of the probability flow ODE. It is worth noting that the corrector steps proposed therein are based on Langevin-type diffusion and inject additive noise, and hence the resulting sampling processes are not deterministic. Additionally, theoretical justifications for DDPM in the context of image in-painting have been developed by Rout et al. (2023). Moreover, convergence results based on the Wasserstein distance have recently been derived as well (Tang, 2023; Benton et al., 2023b), although these results typically exhibit exponential dependency on the Lipschitz constants of the score functions. Theoretical guarantees are also extended to accommodate popular methods like consistency models and diffusion guidance (Li et al., 2024b; Wu et al., 2024).

**Score matching.** Hyvärinen (2005) showed that the score function can be estimated via integration by parts, a result that was further extended in Hyvärinen (2007). Song et al. (2020b) proposed sliced score matching to tame the computational complexity in high dimension. The consistency of the score matching estimator was studied in Hyvärinen (2005), with asymptotic normality established in Forbes & Lauritzen (2015). Optimizing the score matching loss has been shown to be intimately connected to minimizing upper bounds on the Kullback-Leibler divergence (Song et al., 2021a) and Wasserstein distance (Kwon et al., 2022) between the generated distribution and the target data distribution. From a non-asymptotic perspective, Koehler et al. (2023) studied the statistical efficiency of score matching by connecting it with the isoperimetric properties of the distribution.

**Other theory for diffusion models.** Oko et al. (2023) studied the approximation and generalization capabilities of diffusion modeling for distribution estimation. Assuming that the data are supported on a low-dimensional linear subspace, Chen et al. (2023a) developed a sample complexity bound for diffusion models. Moreover, Ghimire et al. (2023) adopted a geometric perspective and showed that the forward and backward processes of diffusion models are essentially Wasserstein gradient flows. Recently, the idea of stochastic localization, which is closely related to diffusion models, is adopted to sample from posterior distributions (Montanari & Wu, 2023; El Alaoui et al., 2022), which has been implemented using approximate message passing (Donoho et al., 2009; Li & Wei, 2022).

## 5 DISCUSSION

In this paper, we have developed a new suite of non-asymptotic theory for establishing the convergence and faithfulness of diffusion generative modeling, assuming access to reliable estimates of the (Stein) score functions. Our analysis framework seeks to track the dynamics of the reverse process directly using elementary tools, which eliminates the need to look at the continuous-time limit and invoke the SDE and ODE toolboxes. Only the very minimal assumptions on the target data distribution are imposed. The analysis framework laid out in the current paper might shed light on how to analyze other variants of score-based generative models as well. Moving forward, there are plenty of questions that require in-depth theoretical understanding. For instance, the dimension dependency in our convergence results remains sub-optimal; can we further refine our theory in order to reveal tight dependency in this regard? Can we establish sharp convergence results in terms of the Wasserstein distance, which could sometimes be "closer" to how humans differentiate pictures and might potentially help relax Assumption 2 in the case of deterministic samplers? It would also be of paramount interest to establish end-to-end performance guarantees that take into account both the score learning phase and the sampling phase.

## ACKNOWLEDGEMENTS

Y. Wei is supported in part by the the NSF grants DMS-2147546/2015447, CAREER award DMS-2143215, CCF-2106778, and the Google Research Scholar Award. Y. Chen is supported in part by the Alfred P. Sloan Research Fellowship, the Google Research Scholar Award, the AFOSR grant FA9550-22-1-0198, the ONR grant N00014-22-1-2354, and the NSF grants CCF-2221009 and CCF-1907661. Y. Chi is supported in part by the grants ONR N00014-19-1-2404, NSF CCF-2106778, DMS-2134080 and ECCS-2126634.

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
