# OpenReview forum: "Towards Non-Asymptotic Convergence for Diffusion-Based Generative Models"
_ICLR.cc/2024/Conference — ICLR 2024 poster_

### Official Review · Reviewer_nMDV · 2023-10-29

**Soundness:** 3 good
**Presentation:** 3 good
**Contribution:** 3 good
**Rating:** 8
**Confidence:** 2

**Summary:**

This paper develops theoretical analyses for two variants of score-based generative models, those with deterministic (known as the probability flow ODE) and stochastic (known as denoising diffusion probabilistic models) reverse time processes, respectively. The deterministic variant is of particular interest because it has been difficult to analyze with existing techniques, despite being successful in practice. The main contribution of this work is to derive convergence guarantees for each of these processes under mild assumptions on the data distribution and the score function. Briefly, they provide an analysis of the deterministic algorithm which is the first to provide explicit rates of convergence, when the data distribution has bounded support (with only logarithmic dependence on the diameter) and when the score function and its Jacobian have been estimated. For the stochastic algorithm they recover similar results to the state-of-the-art, albeit with worse dimension dependence and stronger assumptions on the data distribution (compact support vs finite second moment).

**Strengths:**

This work studies an important problem, that of developing theoretical understanding of the efficacy of score-based methods, and focuses in particular on deterministic methods, which are not well understood theoretically. They develop the first analysis to achieve explicit rates of convergence for these methods in the literature, and achieve a strong bound under assumptions which are quite mild (except for the Jacobian assumption). For the stochastic method, they gain results which are close to the best known. Their analysis is necessarily novel and avoids technical difficulties that have forced previous works [1] to study (stochastic) modifications of the fully deterministic methods.

[1] The Probability Flow ODE Is Provably Fast, by Chen et al 2023

**Weaknesses:**

From my point of view, the main weakness of the results is that they involve the error of the differential of the score function, and it is not clear why this would be well-controlled in general (since the score-matching objective doesn't involve the Jacobian). The prior work [2] didn't use such conditions, but then again achieved far weaker guarantees. It would be good to comment on the importance and plausibility of this condition. Less significant but nonetheless important is the authors' use of very specific step-size schemes, a remark on this would also be helpful. Finally, their convergence results are proved in TV (weaker than KL) and are not for the true data distribution, but for the distribution of the first step of the forward diffusion process.

With regard to correctness of their arguments, I was not able to carefully check this point due to the long and involved natured of their proofs but everything that I did read in the supplementary material seems correct.

[2] "Restoration-degradation beyond linear diffusions: A non-asymptotic analysis for DDIM-type samplers" by Chen, Daras, and Dimakis 2023

**Questions:**

- Please add a remark about the fact that you prove your guarantees for convergence to the first step of the forward process (rather than the true data distribution). Of course, this is what allows for guarantees in TV with minimal assumptions on the data distribution since otherwise the data distribution could be singular, but I wonder how much this changes your results.
- Are the step-size schemes you consider similar to those used in practice? How robust are your results to variations in the step-sizes?
- Please discuss the utility and plausibility of the bound on the difference of Jacobians
- Could you please clarify the meaning of "continuous" in the assumption on the data distribution when it is says "$X_0$ is a continuous random vector, and". In particular, are you assuming the data distribution has a density wrt Lebesgue here?

---

> ### Author Response · Authors · 2023-11-22
>
> Thanks much for the valuable feedback!  Below is a point-by-point response to the reviewer's main comments.
>
> **About the necessity of Jacobian error assumptions.**
>
> Thanks for your comments regarding the assumption on the Jacobian errors. Note, however, that for deterministic samplers, having a small score estimation error alone is insufficient to guarantee convergence when measured by either the TV distance or the KL divergence; in other words, having additional assumptions on the Jacobian estimation error is unavoidable for the deterministic sampler in the presence of score errors. In comparison, [2] does not require this assumption mainly because they assume that the score estimation is noise-free.
>
> In order to rigorize this crucial observation, we have come up with a hard instance as follows:
>
> **A hard example.** Consider the case where $X_{0}\sim\mathcal{N}(0,1)$, and hence $X_{1}\sim\mathcal{N}(0,1)$. Suppose that the reverse process for time $t=2$ can lead to the desired distribution if exact score function is employed, namely,
> $$Y_{1}^{\star}\coloneqq\frac{1}{\sqrt{\alpha_{2}}}\left(Y_{2}-\frac{1-\alpha_{2}}{2}s_{2}^{\star}(Y_{2})\right)\sim\mathcal{N}(0,1). $$
> Now, suppose that the score estimate $s_{2}(\cdot)$ we have available obeys
> $$s_{2}(y_{2})=s_{2}^{\star}(y_{2})+\frac{2\sqrt{\alpha_{2}}}{1-\alpha_{2}}\left\\{ y_{1}^{\star}-L\left\lfloor \frac{y_{1}^{\star}}{L}\right\rfloor \right\\}
> 		\qquad \text{with } y_{1}^{\star}\coloneqq\frac{1}{\sqrt{\alpha_{2}}}\left(y_{2}-\frac{1-\alpha_{2}}{2}s_{2}^{\star}(y_{2})\right)$$
> for some large integer $L>0$, where $\lfloor z \rfloor$ is the greatest  integer not exceeding $z$. It follows that
> $$Y_{1}=Y_{1}^{\star}+\frac{1-\alpha_{2}}{2\sqrt{\alpha_{2}}}\big[ s_{2}^{\star}(Y_{2})-s_{2}(Y_{2})\big]  =L\left\lfloor \frac{Y_{1}^{\star}}{L}\right\rfloor .$$
> Clearly, the score estimation error $\mathbb{E}\_{X_2\sim \mathcal{N}(0,1)}\big[|s_{2}(X_{2})-s_{2}^{\star}(X_{2})|^2\big]$
> can be made arbitrarily small by taking $L$ to be sufficiently large. However, the discrete nature of $Y_{1}$ forces the TV distance to be
> $$\mathsf{TV}(Y_{1},X_{1}) = 1. $$
> This example demonstrates that, for the deterministic sampler, the TV distance between $Y_1$ and $X_1$ might not improve as the score error decreases, if only the score estimation error assumption is imposed. This is in stark contrast to the stochastic sampler. If we wish to eliminate the need of imposing Assumption 2 (about the Jacobian error),
> one potential way is to resort to other metrics (e.g., the Wasserstein distance) instead of the TV distance between $Y_1$ and $X_1$.
>
> We have added this hard example in Section 3 of the revised paper, in the hope of addressing the reviewer's concern.
>
> **About stepsize schemes.**
>
> Thanks for your question regarding our choices of the learning rates. There are several remarks that we would like to make regarding the learning rate schedule.
>
> - It is worth noting that our analysis can be easily extended to accommodate a much broader class of learning rates, and the same convergence behavior can be achieved if the properties (43) are satisfied. If these properties are not satisfied,  our analysis framework is oftentimes still applicable, although the resulting convergence rates might be slower.
>
> - At a more technical level regarding our learning rate designs, we note that in general, the discretization error depends crucially upon the quantity $\frac{1-\alpha_t}{1-\overline{\alpha}_t}$, whereas the initialization error relies heavily upon $\overline{\alpha}_1$ and $\overline{\alpha}_T$. Based on these observations, our learning rate schedule (22) is designed to make $\frac{1-\alpha_t}{1-\overline{\alpha}_t}$ as small as possible, while  making sure $\overline{\alpha}_1$ (resp. $\overline{\alpha}_T$) is close to $1$ (resp. $0$). These properties are shown in (43).
>
> We have added remarks to Section 3.1 of the revised version accordingly to address this comment.
>
> **About convergence to the first step of the forward process.**
>
> Indeed, our results are concerned with the first step of the forward process. Fortunately, given that $X_1 \sim q_1$ and $X_0 \sim q_0$ are quite close through taking $\alpha_1 = 1 - \frac{1}{\mathsf{poly}(T)}$, focusing on the convergence w.r.t. $q_1$ instead of $q_0$ remains practically relevant while at the same time enabling a more effective theory. We have added a remark in the revised paper to explain this point.
>
> **About ''continuous'' random vectors.**
>
> Thanks for raising this concern, and what we mean by "continuous" is exactly what the reviewer describes. We have changed it to "absolutely continuous" in the revised paper, meaning that there exists a probability density w.r.t. the Lebesgue measure.

---

> > ### Comment · Reviewer_nMDV · 2023-11-22
> > **Thanks for your detailed response**
> >
> > Thanks for your thoughtful and detailed response. I am satisfied with your additions and remarks other than those related to the Jacobian assumption.
> >
> > For the Jacobian assumption, the "hard example" looks quite promising, but I am not clear why $\mathbb{E}\_{x_2 \sim \mathcal{N}(0, 1)}[ |s\_2(x\_2) - s\_2^\star(x\_2)|^2]$ is small as $L \to \infty$? It seems that for each fixed $x\_2$,
> > as $L \to \infty$ the term $x\_1^\star - L\lfloor \frac{x\_1^\star}{L} \rfloor \to x\_1^\star \neq 0$?

---

> > > ### Author Response · Authors · 2023-11-22
> > >
> > > Thanks a lot for pointing out this issue. We are sorry about this typo. Here, we should take $L \to 0$, such that $Y_1$ is a discrete variable with $\mathsf{TV}(Y_{1},X_{1}) = 1$, and meanwhile $\mathbb{E}\_{X_2\sim \mathcal{N}(0,1)}\big[|s_{2}(X_{2})-s_{2}^{\star}(X_{2})|^2\big] \asymp \mathbb{E}\Big[\left\\{ X_{1}^{\star}-L\left\lfloor \frac{X_{1}^{\star}}{L}\right\rfloor \right\\}^2\Big] \to 0$. We have corrected this issue in the revised version.

---

### Official Review · Reviewer_h2T8 · 2023-10-30

**Soundness:** 3 good
**Presentation:** 2 fair
**Contribution:** 2 fair
**Rating:** 8
**Confidence:** 4

**Summary:**

This paper studies the deterministic probability flow ODE-based sampler often used in practice for score-based diffusion. It shows that the ODE-based sampler gets a $1/T$ convergence rate, improving upon the SDE-based sampling rate of $1/\sqrt{T}$. The techniques are elementary, and do not rely on Girsanov's theorem and other techniques from the SDE/ODE toolboxes. While there was prior work (Chen et. al. 2023b) that obtains an improved guarantee for the ODE-based sampler, their analysis required the use of stochastic "corrector steps", while in practice, even just the ODE-based sampler without corrector steps seems to perform well. This is the first work that provides theoretical evidence that the vanilla ODE-based sampler can outperform the SDE-based sampler in practice.

**Strengths:**

- Provides the first analysis of the (vanilla) ODE-based sampler for score-based diffusion models that gives some theoretical evidence for why it outperforms the SDE-based sampler in practice ($1/T$ convergence instead of $1/\sqrt{T}$)
- New analysis that doesn't make use of Girsanov's theorem/other results from the ODE/SDE literature.
- Doesn't require Lipschitzness of score unlike (Chen et al 2023b), but pays in d dependence

**Weaknesses:**

- $d$ dependence is worse than (Chen et al 2023b). In particular, Chen gets a $\sqrt{d}$ dependence for the ODE-based sampler when using stochastic corrector steps, which is better than the previous bound of $d$ for the SDE-based sampler. This paper on the other hand gets a $d^3$ dependence, which is significantly worse than both these bounds.
- Requires Jacobian of score to be estimated accurately, rather than just the score
- Requires the distribution $q_0$ to be bounded, and bound is stated in terms of this bound. In contrast, some of the prior works only required the second moment of $q_0$ to be bounded.
- While new analysis is "elementary" in that it doesn't require SDE/ODE machinery, it seems much longer/more complicated than the previous analyses. Would really appreciate a condensation/proof overview in the main paper.

**Questions:**

- Is it clear that $1/\sqrt{T}$ is tight for the SDE-based sampler, and $1/T$ is impossible?
- What are the barriers to getting a $d$ or $\sqrt{d}$ dependence? Can you write something about this in the main paper?
- Can you include a proof overview in the main paper?
- Why is the proof so long? Can it be condensed?

---

> ### Author Response · Authors · 2023-11-22
>
> Thanks for the valuable questions. Below is a point-by-point response to several remarks of the reviewer's.
>
> **About $d$ dependency.**
>
> We agree that the dependency of $d$ in our results is likely loose (to be more precise, our result shows that the deterministic sampler yields $\varepsilon$-accuracy as long as the iteration complexity exceeds $\widetilde{O}\big( \frac{d^2}{\varepsilon} + \frac{d^3}{\sqrt{\varepsilon}}\big)$, or $\widetilde{O}\big( \frac{d^2}{\varepsilon} \big)$ for small enough $\varepsilon$).
> We would like to make two remarks.
> - On the one hand, the reviewer is absolutely correct that the abovementioned $d$-dependency is worse compared to Chen et al 2023b, the latter of which introduced a stochastic correction step to facilitate analysis.
>
> - On the other hand, the main purpose of our theory (for the deterministic sampler part) is to help establish non-asymptotic convergence guarantees for the probability flow ODE approach. Understanding such pure deterministic samplers is of fundamental importance, as this approach has been shown to be highly effective in practice	and it also sheds light on the design of faster methods (e.g., the DPM solver, and the consistency model). However, existing theory for pure deterministic samplers was highly inadequate; for instance, the only non-asymptotic result before us was Chen et al 2023c, which did not provide concrete polynomial dependency and involved terms that scale exponentially in the Lipschitz constant of the score function. Our main contributions are therefore to establish improved convergence theory for the probability flow ODE approach. In comparison, while adding the stochastic correction step enables improved $d$-dependency (from the theoretical perspective), the resulting sampler studied in  Chen et al 2023b deviates from the deterministic sampler used in practice and hence falls short of explaining the effectiveness of this popular approach.
>
> **About Jacobian errors.**
>
> Thanks for pointing out the additional requirement on the Jacobian estimation accuracy. Note, however, that for deterministic samplers, having a small score estimation error alone is insufficient to guarantee convergence when measured by either the TV distance or the KL divergence; in other words, having additional assumptions on the Jacobian estimation error is unavoidable for the deterministic case. In order to rigorize this crucial observation, we have come up with a hard instance as follows:
>
> **A hard example.** Consider the case where $X_{0}\sim\mathcal{N}(0,1)$, and hence $X_{1}\sim\mathcal{N}(0,1)$. Suppose that the reverse process for time $t=2$ can lead to the desired distribution if exact score function is employed, namely,
> $$Y_{1}^{\star}\coloneqq\frac{1}{\sqrt{\alpha_{2}}}\left(Y_{2}-\frac{1-\alpha_{2}}{2}s_{2}^{\star}(Y_{2})\right)\sim\mathcal{N}(0,1).$$
> Now, suppose that the score estimate $s_{2}(\cdot)$ we have available obeys
> $$s_{2}(y_{2})=s_{2}^{\star}(y_{2})+\frac{2\sqrt{\alpha_{2}}}{1-\alpha_{2}}\left\\{ y_{1}^{\star}-L\left\lfloor \frac{y_{1}^{\star}}{L}\right\rfloor \right\\}
> 		\qquad \text{with } y_{1}^{\star}\coloneqq\frac{1}{\sqrt{\alpha_{2}}}\left(y_{2}-\frac{1-\alpha_{2}}{2}s_{2}^{\star}(y_{2})\right)$$
> for $L>0$, where $\lfloor z \rfloor$ is the greatest  integer not exceeding $z$. It follows that
> $$Y_{1}=Y_{1}^{\star}+\frac{1-\alpha_{2}}{2\sqrt{\alpha_{2}}}\big[ s_{2}^{\star}(Y_{2})-s_{2}(Y_{2})\big]  =L\left\lfloor \frac{Y_{1}^{\star}}{L}\right\rfloor .$$
> Clearly, the score estimation error $\mathbb{E}\_{X_2\sim \mathcal{N}(0,1)}\big[|s_{2}(X_{2})-s_{2}^{\star}(X_{2})|^2\big]$ can be made arbitrarily small by taking $L \to 0$. However, the discrete nature of $Y_{1}$ forces the TV distance to be
> $\mathsf{TV}(Y_{1},X_{1}) = 1. $
> This example demonstrates that, for the deterministic sampler, the TV distance between $Y_1$ and $X_1$ might not improve as the score error decreases, if only the score estimation error assumption is imposed. This is in stark contrast to the stochastic sampler. If we wish to eliminate the need of imposing Assumption 2 (about the Jacobian error),
> one potential way is to resort to other metrics (e.g., the Wasserstein distance) instead of the TV distance between $Y_1$ and $X_1$.
>
> We have added this hard example in Section 3 of the revised paper, in the hope of addressing the reviewer's concern.

---

> > ### Author Response · Authors · 2023-11-22
> >
> > **About the boundedness assumption.**
> >
> > We agree that this requirement is stronger than the bounded second moment. There are two comments that we would like to make here:
> > - It is noteworthy that our result only has a logarithmic dependency on the maximum size of the data, meaning that the convergence rate changes extremely slowly as the maximum size of $X_0$ increases. More specifically, using the same analysis as for Theorem 1, we can straightforwardly derive
> > \begin{align}
> > \mathsf{TV}\big(q_1, p_1\big) \leq C_1 \frac{d^2\log^4 T\log^2 B}{T} + C_1\frac{d^{6}\log^{6} T\log^3 B}{T^2}
> > 	+C_1\sqrt{d\log^{3}T\log B}\varepsilon_{\mathrm{score}}+ C_1d(\log T)\varepsilon_{\mathrm{Jacobi}}
> > \end{align}
> > where we use $B$ to denote the maximum value of $\\|X_0\\|_2$.
> >
> > - In addition, in most applications of the diffusion model, the data (e.g., the image data) is supported on a compact set. In fact, the model is trained based on the empirical dataset, where the data is naturally bounded by some finite (although possibly large) quantity. In light of this, we believe that our result remains consistent with practical constraints, given that our current convergence rate remains unchanged even when the size of the true data scales polynomially in $T$ (which can be polynomials with arbitrarily large degrees).
> >
> > We have added a remark in the revised paper to clarify these points.
> >
> > **About the proof overview.**
> >
> > Thanks a lot for your suggestion.
> > We agree that including a subsection detailing the proof overview would be quite beneficial.
> > Unfortunately, mainly due to the space limitations, the main analysis steps of Theorems 1-2 have to be deferred to the supplementary material.
> > To address the reviewer's comment,
> > we have now added a new section in the supplementary material (see Appendix A of the revised paper) to provide a proof outline.
> >
> > **About SDE rate.**
> >
> > So far, all existing theory for the vanilla SDE approach (i.e., the DDPM approach) exhibits a convergence rate that is no faster than $1/\sqrt{T}$.  It might be possible to further accelerate it via proper modification of the algorithm (e.g., exploiting certain momentum term). Showing the fundamental limit of the stochastic approach is a problem of fundamental importance, which we leave for future studies.
> >
> > **Technical difficulty regarding sub-optimal $d$-dependency.**
> >
> > The suboptimal dependency on $d$ in our theory comes mainly from Lemma 4 (which has been used to establish our main result). The main difficulties to improve Lemma 4 lie in the calculations of some quantities regarding the conditional distribution of $x_0$ given $x_t$, for example, the Jacobian matrix of $s_t$. We have added a remark to explain this point in our final version.

---

> ### Comment · Reviewer_h2T8 · 2023-11-23
>
> Thank you for your response. I appreciate the inclusion of the hard example and the proof overview.
>
> What kind of guarantee would you be able to achieve for sampling with Wasserstein closeness, if you don't have the Jacobian guarantee? Would really appreciate it if you could include this result as well.
>
> I will raise my score to 8 regardless, but I would really appreciate it if this result appeared in the final version.

---

> > ### Author Response · Authors · 2023-11-23
> >
> > Thank you a lot for your insightful feedback, and for increasing your score for our paper.
> >
> > Regarding the Wasserstein-distance-based results, our current conjecture is that: the probability flow ODE sampler might achieve
> > $W_2(q_1,p_1)\lesssim \mathsf{poly}(d) \big( \frac{1}{T} + \varepsilon_{\sf score} \big)$,
> > provided that Assumption 1 and (21) hold, which does not involve Assumption 2 (the assumption concerning the Jacobian error). Nevertheless, while our analysis framework can be adapted to tackle the Wasserstein distance,
> > our current analysis encounters some obstacle regarding the dependency on certain Lipschitz constants of the score estimates (given that the Jacobian error bound is not assumed).  For instance, we are in  need of  controlling how the associated estimation error of $y_t$ propagates to $y_1$,
> > and so far existing theory all exhibits exponential dependency on the Lipschitz constants.
> > We are currently trying our best to find solutions to cope with this issue, and will include the result in the final version if we can resolve this fundamental issue satisfactorily.

---

### Official Review · Reviewer_ScYQ · 2023-10-31

**Soundness:** 4 excellent
**Presentation:** 3 good
**Contribution:** 4 excellent
**Rating:** 8
**Confidence:** 5

**Summary:**

This paper aims to provide a systematic analysis of the convergence rate of both deterministic and stochastic samplers of the diffusion models in the context of generative modeling. The authors proves, under certain assumptions, the former has a rate of $T^{-1}$
while the latter has a rate $\sqrt{T}$ which is consistent with the previous empirical observations. The authors also proposed some improvements, leading to a rate of $T^{-2}$ in the deterministic case, and a rate of $T^{-1}$ in the stochastic case.

**Strengths:**

This paper provides an early systematic analysis on the convergence rate of samplers in the diffusion models. Both deterministic and stochastic cases are considered, and their results are almost optimal under the assumptions made. The presentation of the paper is good, and I really enjoyed reading it. Especially, I like Theorem 1.

**Weaknesses:**

As for every (good) paper, there are always plenty of things remained to be done. For instance,

(1) It is worthy to comment on the learning rate (22a)-(22b). I understand these rates are carefully chosen in order to match the rates in the theorems. The author may mention this, or provide some explanations/insights on it.

(2) The paper, like Chen et al., deals with the TV (or KL) divergence. I understand that under these metrics, one can prove "nice" theoretical results using some specific algebraic identities (flow...) On the other hand, practitioners may care more about the FID (or Wasserstein distance) -- part of the reason is that Wasserstein distance is "closer" to how humans distinguish pictures. The authors may want to add a few remarks on this.

**Questions:**

See the weaknesses.

---

> ### Author Response · Authors · 2023-11-22
>
> We thank the reviewer for the positive appraisal of our paper and for helpful feedback! Please find below our point-by-point response to some of the main comments.
>
> **About the learning rates.**
>
> Thanks for suggesting more explanations about our choices of the learning rates. There are several remarks that we would like to make regarding the learning rate schedule.
> - As can be easily verified, in general the discretization error depends crucially upon the quantity $\frac{1-\alpha_t}{1-\overline{\alpha}_t}$, whereas the initialization error relies heavily upon $\overline{\alpha}_1$ and $\overline{\alpha}_T$.
> Based on these observations, our learning rate schedule (22) is designed to make $\frac{1-\alpha_t}{1-\overline{\alpha}_t}$ as small as possible, while  making sure $\overline{\alpha}_1$ (resp. $\overline{\alpha}_T$) is close to $1$ (resp. $0$). These properties are shown in (43).
>
> - On the other hand, it is worth noting that our analysis can be easily extended to accommodate a much broader class of learning rates, and the same convergence behavior can be achieved if the properties (43) are satisfied. If these properties are not satisfied,  our analysis framework is oftentimes still applicable, although the resulting convergence rates might be slower.
>
> We have added remarks to Section 3.1 of the revised version accordingly to address this comment.
>
> **About TV/KL vs. the Wasserstein distance.**
>
> Thanks a lot for your suggestion, and we fully agree with your comments regarding the practical importance of the Wasserstein distance as a performance metric. There are several remarks that we would like to make in this regard.
> - In fact, there have been several recent papers establishing appealing convergence guarantees based on the Wasserstein distance (e.g., Tang 2023, and Benton et al., 2023). One weakness of these Wasserstein-distance-based results,
> however, is that the iteration complexities of these results exhibit exponential dependency on the Lipstchiz constant of the score function. This is in stark contrast to our result and Chen et al., as these TV/KL-based convergence rates are nearly independent of the Lipschitz constant of the score function.
>
> - For the deterministic sampler, our convergence results rely upon Assumption 2, which is concerned with the accuracy of Jacobian matrices. In Section 3 of the revised paper, we have included a hard example illustrating the necessity of such an assumption. However, we conjecture that Assumption 2 might be non-essential if the performance metric is chosen to be the Wasserstein distance (given that the Wasserstein distance allows one to transport probability mass in order to mitigate the mismatch between continuous and discrete random variables). In light of this conjecture, it would be valuable to develop a non-asymptotic convergence result for the deterministic sampler based on the Wasserstein distance.
>
> We have added some remarks regarding the Wasserstein-distance-type results in the revised paper.
>
> **References:**
> - Wenpin Tang. Diffusion Probabilistic Models. (2023).
> - Joe Benton, George Deligiannidis, and Arnaud Doucet. Error bounds for flow matching methods. arXiv preprint arXiv:2305.16860, 2023.

---

### Official Review · Reviewer_TgkF · 2023-11-01

**Soundness:** 4 excellent
**Presentation:** 3 good
**Contribution:** 3 good
**Rating:** 6
**Confidence:** 3

**Summary:**

The paper delves into the intricacies of diffusion models, a unique class of models capable of converting noise into fresh data instances through the reversal of a Markov diffusion process. While the practical applications and capabilities of these models are well-acknowledged, there remains a gap in the comprehensive theoretical understanding of their workings. Addressing this, the authors introduce a novel non-asymptotic theory, specifically tailored to grasp the data generation mechanisms of diffusion models in a discrete-time setting. A significant contribution of their research is the establishment of convergence rates for both deterministic and stochastic samplers, given a score estimation oracle. The study underscores the importance of score estimation accuracies. Notably, this research stands apart from prior works by adopting a non-asymptotic approach, eschewing the traditional reliance on toolboxes designed for SDEs and ODEs.

**Strengths:**

This paper offers a significant theoretical advancement by providing the convergence bounds for both stochastic and deterministic samplers of diffusion processes. I'm particularly struck by the elegance of their results, especially given the minimal assumptions required—for instance, the results for stochastic samplers rely solely on Assumption 1. These theoretical findings provide a clear understanding of the effects of score estimation inaccuracies. Furthermore, the emphasis on the estimation accuracies of Jacobian matrices for deterministic samplers sheds light on potential training strategies, suggesting the incorporation of penalties in objective functions when learning the score function for these samplers. Additionally, the proof presented is elementary, and potentially useful in a broader context.

**Weaknesses:**

- While this paper doesn't present any algorithmic advancements, it's understandable considering the depth of their theoretical contributions.

- The proof appears to be procedural. I was hoping for deeper insights into how the proof was constructed. It might be beneficial for the authors to include a subsection detailing the outline and insights behind their proof.

**Questions:**

- Eq. (14), just want to confirm: should it be $dX_t = \sqrt{1-\beta_t}X_td_t$ instead of $-\frac{1}{2}\beta(t)X_t d_t$?
- Eq. (15), maybe I am missing something, but should it be $dY = (-f(...) + \frac{1}{2}g^2 \nabla) d_t$? The original paper was taking a reverse form.
- Eq. (16), similar question to Eq. (15).
- Eq. (21), I feel a bit weird about the notation: maybe remove the $R$.
- The results for deterministic samplers may motivate a better training strategy (i.e., incorporating the requirement for Jacobian matrices accuracy). The first step is to verify that this phenomenon exists in experiments (i.e., a worse $\epsilon_{Jacobi}$ does imply a worse sampling result). It may be worth adding a numerical experiment to confirm this if time permits.

---

> ### Author Response · Authors · 2023-11-22
>
> Thanks a lot for the helpful comments and valuable feedback.
> Please find below our point-by-point response to the main comments, which has been incorporated into the revision version.
>
> **About proof outline.**
> We agree that including a subsection detailing the proof outline would be quite beneficial.
> Currently, mainly due to the space limitations, the main analysis steps of Theorems 1-2 are deferred to the supplementary material.
> We have now added a new section in the supplementary material (see Appendix A of the revised paper) to provide a proof outline.
>
> **About Eq. (14).**
> Thanks for pointing out the confusion.
> Note that the continuous-time limit of (4) can be understood in the sense that
> $$X_{t}-X_{t-\mathrm{d}t}=\sqrt{1-\beta_{t}}X_{t-\mathrm{d}t}-X_{t-\mathrm{d}t}+\sqrt{\beta_{t}}W_{t}\approx-\frac{1}{2}\beta_{t}X_{t-\mathrm{d}t}+\sqrt{\beta_{t}}W_{t},$$
>
> provided that $\beta_t$ is small; this is why the continuous-time limit reads
> $$\mathrm{d}X_{t}=-\frac{1}{2}\beta(t)X_{t}+\sqrt{\beta_{t}}\mathrm{d}W_{t}.$$
> We have added a footnote in the revised paper to make this more clear.
>
> **About Eqs. (15) and (16).**
> Thanks for pointing the typos!  Indeed, these should be $-f$ rather than $f$. We have fixed them in the revised version.
>
> **About Eq. (21).** Thanks for pointing out this notational issue. We have removed $R$ in the revised version as suggested by the reviewer.
>
> **About potential training strategies with Jacobian requirements.**
> Thanks a lot for the suggestion! Indeed, the additional requirement about the Jacobians in our theory motivates us to explore potential strategies to enforce such requirements in the training phase.
>
> After careful examination of this requirement, we realize that
> - The additional Jacobian requirement is largely due to the use of the TV distance (or KL divergence) as a performance metric; in other words, if only the score estimation error is assumed, then one can come up with a hard example such that the TV distance remains large even when the number of steps $T$ is large. We shall also include the hard example below, which is largely due to the mismatch between continuous random variables and discrete random variables.
>
> - However, we conjecture that this fundamental requirement on Jacobian accuracy might be non-essential if the performance metric is chosen to be the Wasserstein distance (given that the Wasserstein distance allows one to transport probability mass in order to mitigate the mismatch between continuous and discrete random variables). If our conjecture were true, then it would be unclear whether enforcing the additional Jacobian estimation accuracy would be beneficial.
>
>  We will conduct some experiments to test the utility of enforcing such Jacobian-based requirements in the training phase.

---

> ### Author Response · Authors · 2023-11-22
> **The hard example**
>
> The hard example mentioned above is described below, which has also been incorporated into Section 3 of the revised paper.
>
> Consider the case where $X_{0}\sim\mathcal{N}(0,1)$, and hence $X_{1}\sim\mathcal{N}(0,1)$. Suppose that the reverse process for time $t=2$ can lead to the desired distribution if exact score function is employed, namely,
> $$Y_{1}^{\star}\coloneqq\frac{1}{\sqrt{\alpha_{2}}}\left(Y_{2}-\frac{1-\alpha_{2}}{2}s_{2}^{\star}(Y_{2})\right)\sim\mathcal{N}(0,1).$$
> Now, suppose that the score estimate $s_{2}(\cdot)$ we have available obeys
> $$s_{2}(y_{2})=s_{2}^{\star}(y_{2})+\frac{2\sqrt{\alpha_{2}}}{1-\alpha_{2}}\left\\{ y_{1}^{\star}-L\left\lfloor \frac{y_{1}^{\star}}{L}\right\rfloor \right\\}
> 		\qquad \text{with } y_{1}^{\star}\coloneqq\frac{1}{\sqrt{\alpha_{2}}}\left(y_{2}-\frac{1-\alpha_{2}}{2}s_{2}^{\star}(y_{2})\right)$$
> for $L>0$, where $\lfloor z \rfloor$ is the greatest  integer not exceeding $z$. It follows that
> $$Y_{1}=Y_{1}^{\star}+\frac{1-\alpha_{2}}{2\sqrt{\alpha_{2}}}\big[ s_{2}^{\star}(Y_{2})-s_{2}(Y_{2})\big]  =L\left\lfloor \frac{Y_{1}^{\star}}{L}\right\rfloor .$$
> Clearly, the score estimation error $\mathbb{E}\_{X_2 \sim \mathcal{N}(0,1)}\big[|s_{2}(X_{2})-s_{2}^{\star}(X_{2})|^2\big]$
> can be made arbitrarily small by taking $L \to 0$.
> However, the discrete nature of $Y_{1}$ forces the TV distance to be
> $$\mathsf{TV}(Y_{1},X_{1}) = 1. $$
> This example demonstrates that, for the deterministic sampler, the TV distance between $Y_1$ and $X_1$ might not improve as the score error decreases, if only the score estimation error assumption is imposed.
> This is in stark contrast to the stochastic sampler.
> If we wish to eliminate the need of imposing Assumption 2 (about the Jacobian error),
> one potential way is to resort to other metrics (e.g., the Wasserstein distance) instead of the TV distance between $Y_1$ and $X_1$.

---

> > ### Comment · Reviewer_TgkF · 2023-11-23
> >
> > This example is interesting as it underscores the importance of the Jacobian matrix requirement. I appreciate the thorough responses to my comments. Thank you!

---

### Meta-Review · Area_Chair_EgdY · 2023-12-09

**Metareview:**

This paper extended the existing analyses on diffusion models to the algorithms with deterministic updates and obtained nonasymptotic convergence guarantees. The reviewers feel that the paper offers enough theoretical insight to justify its acceptance.

The authors should however pay attention to the accuracy of some of the statements. For example, on top of page 9, the authors claim that "for any smooth distribution satisfying the log-Sobelev inequality, effectively only allowing unimodal distributions", which is incorrect. Log-Sobelev inequality does not imply that the distribution is unimodal.

In addition, if the paper can weaken the assumptions on how well the scores are learned and instead provide algorithms for such subroutine with theoretical guarantees, that would make the theoretical contribution complete and warrants an even higher recommendation. There in fact exists works that can take the computation of the score matching step into account (c.f., "reverse diffusion Monte Carlo").

**Justification For Why Not Higher Score:**

If the paper can weaken the assumptions on how well the scores are learned and instead provide algorithms for such subroutine with theoretical guarantees, that would make the theoretical contribution complete and warrants an even higher score. There in fact exists works that can take the computation of the score matching step into account (c.f., "reverse diffusion Monte Carlo").

**Justification For Why Not Lower Score:**

The reviewers feel that the paper offers enough theoretical insight to justify its acceptance.

---

### Decision · Program_Chairs · 2024-01-16

Accept (poster)